# Improving Small-Scale Human Action Recognition Performance Using a 3D Heatmap Volume

**DOI:** 10.3390/s23146364

**Published:** 2023-07-13

**Authors:** Lin Yuan, Zhen He, Qiang Wang, Leiyang Xu, Xiang Ma

**Affiliations:** Department of Control Science and Engineering, Harbin Institute of Technology, Harbin 150001, China; hezhen@hit.edu.cn (Z.H.); wangqiang@hit.edu.cn (Q.W.); xuleiyang@stu.hit.edu.cn (L.X.); 15b904026@hit.edu.cn (X.M.)

**Keywords:** fine-grained action recognition, small-scale dataset, Tai Chi action, heatmap volume

## Abstract

In recent years, skeleton-based human action recognition has garnered significant research attention, with proposed recognition or segmentation methods typically validated on large-scale coarse-grained action datasets. However, there remains a lack of research on the recognition of small-scale fine-grained human actions using deep learning methods, which have greater practical significance. To address this gap, we propose a novel approach based on heatmap-based pseudo videos and a unified, general model applicable to all modality datasets. Leveraging anthropometric kinematics as prior information, we extract common human motion features among datasets through an ad hoc pre-trained model. To overcome joint mismatch issues, we partition the human skeleton into five parts, a simple yet effective technique for information sharing. Our approach is evaluated on two datasets, including the public Nursing Activities and our self-built Tai Chi Action dataset. Results from linear evaluation protocol and fine-tuned evaluation demonstrate that our pre-trained model effectively captures common motion features among human actions and achieves steady and precise accuracy across all training settings, while mitigating network overfitting. Notably, our model outperforms state-of-the-art models in recognition accuracy when fusing joint and limb modality features along the channel dimension.

## 1. Introduction

The field of human action recognition has seen extensive research efforts, resulting in the establishment of several human action datasets with various data modalities, including RGB videos, depth, radar [1], infrared radiation (IR) and skeleton coordinates [2]. Prevailing research focuses on the utilization of deep and complex neural networks to achieve impressive outcomes, and this is supported by the construction of large-scale human action datasets, such as UCF-101 [3] with 13 K samples and Kinetics-600 [4] with 392 K samples. However, the potential of these extensive multi-modal human action datasets should not be limited to verifying the performance of these algorithms. Instead, it is imperative to extensively explore their application value, such as for aiding the recognition of small-scale fine-grained datasets.

The present human action datasets typically focus on coarse-grained daily activities, exhibiting noticeable inter-class differences and featuring major movements from only certain body parts. Conversely, studies of specific fine-grained actions hold significant practical value, as they can be considered as a general action within a coarse-grained setting. For instance, dance can include ballet, salsa, and tango, among others. A highly effective model can identify subtle nuances and provide appropriate feedback in interactive tasks as well as assess athletic performance in action evaluation scenarios. However, due to limited sampling conditions or high sampling costs, the number of samples in these types of datasets is significantly fewer than in mainstream datasets. Therefore, the recognition of fine-grained actions using deep learning methods with limited training samples remains a challenging and noteworthy endeavour, which constitutes the primary contribution of this paper.

Following the creation of these datasets, numerous related methods have been proposed for human action recognition tasks. Early methods relied on hand-crafted features to differentiate between different action classes. However, with the emergence of deep learning, various methods such as 2D convolutional neural networks (2D-CNN), long short-term memory (LSTM), 3D-CNN, graph convolutional network (GCN), and Transformer have been employed to recognize samples, and classification accuracy has steadily improved. Currently, when performing skeleton-based human action recognition, researchers consider GCN or Transformer as essential tools, whereas 3D-CNN and TimeSformer [5] are popular in video human action recognition.

The aforementioned methods exhibit both advantages and limitations regarding specific sample modalities. Video modality samples cannot be effectively represented by the graph node representation of GCN, whereas skeleton data cannot be directly utilized as input for CNNs. This demarcation presents a natural boundary between video-based and skeleton-based actions. In the case of skeleton modality action samples, local joints may be absent, and multi-person interactive actions may occur. Previous studies utilizing GCN-based methods addressed such situations by performing linear interpolation based on the remaining joints to complete the skeleton structure [6] and taking the average of each performer’s output as the action feature [7]. However, the supplemented skeleton structure may contain unreliable information and introduce interference information. Additionally, averaging the features of multi-person ignores interactive information between characters. If only small-scale fine-grained datasets are used to train deep neural networks from scratch, the network is prone to over-fitting. Prior studies have employed adaptive sharpness aware minimization [8] (ASAM) or pre-trained models using the ImageNet dataset [9] in image processing to address these challenges. Building upon the pre-training strategy, our study endeavours to leverage public action datasets to pre-train the network and tackle the problem of mismatching joints and modalities between different datasets.

In the context of using deep neural networks for small-scale human action recognition, issues such as missing skeleton data, interactive samples, and over-fitting have yet to be addressed. We propose a unified and general model **SSCLS** for **S**mall-**S**cale Action **CL**a**S**sification. Figure 1 shows the network of our recognition framework. Unlike mainstream skeleton-based human action recognition tasks, we introduce **3D heatmap volumes**, also called **heatmap-based pseudo videos**, as network inputs instead of video images or skeleton sequences. Representative heatmap frames, derived from two different formats, are shown in Figure 2. For any frame of the video input, we estimate human 2D poses with a high-resolution network (HRNet) [10]. For original 3D skeleton coordinates, we transform (x,y) 2D coordinates into a heatmap at each time step. In this way, we can obtain a heatmap $h\in R^{C\times H\times W}$, where *C*, *H*, and *W* refer to the channel, height, and width of the heatmap. After calculating the heatmaps of every single frame, we concatenate these heatmaps along the temporal dimension to form a 3D heatmap volume $H\in R^{C\times T\times H\times W}$. As illustrated in Figure 1, the public NTU RGB+D samples [11] have 17-channel inputs for 17 skeleton joints, and our self-built Tai Chi actions [12] have 72 joints. To address the skeleton joint mismatch problem among different datasets while using complete joint information of samples, we divide the skeleton joints into five parts, including four limbs and a trunk, and uniform samples into 5-channel inputs, marked red, green, black, purple, and orange in Figure 1, while ensuring that each channel holds its physical meanings. When taking 3D heatmap volumes as network inputs, we use the mainstream PoseConv3D model [13] as our model backbone. In this work, we use the NTU RGB+D dataset for network pre-training, and small-scale datasets are only used for the final classification head training or fine-tuning the pre-trained network.

We summarize the contributions of this paper as follows:A unified action representation. We introduce heatmap-based pseudo videos to unify the format of action inputs. This format can convert video-based and skeleton-based samples into the uniform inputs, eliminating the obstacles between these formats.An anthropometric kinematics prior. We propose and demonstrate that human actions have common motion features, so a pre-trained model backbone can help extract human features to overcome the network over-fitting and improve recognition performance on small-scale datasets.A uniform joint segmentation method. We divide the skeleton joints into five parts according to the human structure to sufficiently utilize data information among different datasets regardless of the format and sampler numbers of action data.A state-of-the-art performance. We experiment under various configurations separately on the public Nursing Activities and the self-built Tai Chi datasets. Recognition performance and t-SNE [14] visualization of extracted features illustrate that, compared with previous SOTA methods, our recognition accuracy has been improved to a large extent and the over-fitting phenomenon of the network has been significantly alleviated. When we fuse the features extracted from joints and limbs, the recognition accuracy is further improved, which proves the generality of our proposed model. Our experiment records and results are available at website https://github.com/eunseo-v/SSCLS (accessed on 10 May 2023).

## 2. Related Work

This section aims to provide an overview of the current state-of-the-art in skeleton-based human action recognition research, including relevant datasets, recognition algorithms, small-scale fine-grained action datasets, pre-training strategies, and evaluation protocols. We also perform a comparative analysis of our proposed approach with other related work, including prior research on Tai Chi action recognition.

### 2.1. Skeleton-Based Human Action Datasets

This section focuses on skeleton-based human action datasets and their acquisition methods, which can be categorized into three groups: depth-based, wearable-based, and extraction-based. Among them, the Microsoft Kinect system [15] is widely used for 3D human action extraction based on depth sensors. Two large-scale human daily activity datasets, namely the NTU RGB+D dataset [11,16] and the PKU-MMD dataset [17], contain skeleton coordinates collected by the Kinect V2 sensor and can extract multiple human motions using pose estimation algorithms. However, the accuracy of these datasets is affected by factors such as light intensity and background. Another type of dataset is acquired using wearable motion-capture systems, such as the Perception Neuron [18], which utilizes inertial sensors to measure the orientation and acceleration of the wearer. Our self-built Tai Chi action dataset [19] was collected using the Perception Neuron system and consists of only 3D skeleton coordinates. Although this method can accurately capture the skeleton coordinates, it is limited to single sampler action, and the measurement accuracy can be influenced by the drift of inertial sensors.

In the realm of extraction-based datasets, pose estimation algorithms have been widely utilized for 2D human pose detection and estimation from videos, enabling direct acquisition of skeleton coordinates from action video clips. Compared to 3D pose estimators, estimated 2D skeletons are found to be more accurate and robust. Notably, a high-resolution network (HRNet) [10] has been proposed as a means to predict a more precise and spatially accurate keypoint heatmap, and MediaPipe [20] serves as a cross-platform machine learning (ML) framework that can extract human poses from videos, as exemplified by MediaPipe Hands [21], a real-time on-device hand tracking solution that predicts a hand skeleton. Such methods offer a convenient way to adjust the format of network inputs and cost-effectively enrich the human action dataset.

In our study, we utilize the skeleton coordinates of the NTU RGB+D dataset as our pre-trained dataset, which features 60 action classes spanning daily, mutual, and health-related actions captured across three camera views and numerous subjects. Given our supposition that human activities share common features, we anticipate that the pre-trained data will harbour ample action information, and thereby facilitate our small-scale action recognition task.

### 2.2. Methods for Human Action Recognition

As the method section is not the main focus of this paper, we will briefly introduce human action recognition. Previous methods have relied on hand-crafted descriptors to represent human motions, with varying degrees of success. For example, Wang et al. [22] attempted to improve motion-based descriptors by incorporating camera motion into their approach. At the same time, Oreifej and Liu [23] developed a novel descriptor called the histogram of 4D oriented gradients (HON4D), which created 4D projectors of time and space. These descriptors were specifically designed and were less generalizable to all datasets. In recent years, deep learning-based approaches have gained popularity in action recognition research [24]. These approaches have primarily focused on using skeleton-based sequences as a sequence problem, with LSTM-based models [25,26,27,28] being proposed to construct temporal relationships of the sequence with different spatial alignments. Researchers [29,30,31,32] have also explored the use of pseudo-images to represent 3D skeleton sequences, which allows pre-trained mainstream CNN-based networks to be fine-tuned for feature extraction without requiring training from scratch. For instance, Wang and Li [31] proposed transforming 3D skeleton sequences into joint trajectory maps, which were then classified using CNN kernels. However, these approaches have been criticized for breaking down the physical structure of skeleton sequences and lacking spatial and temporal dynamics.

To address these limitations, Kipf et al. [33] developed an efficient variant of convolutional neural networks (CNNs) that operate directly on graphs for graph-structured data. This approach generalizes CNNs to graphs of arbitrary structures, making it well-suited to represent 3D skeleton sequences, which can be naturally regarded as graph data. Yan et al. [34] then proposed a novel model called spatial–temporal graph convolutional networks (ST-GCN), which builds upon Kipf’s method and can automatically learn the spatial configuration of the joints and temporal dynamics. This model has become the baseline comparison for subsequent studies on action recognition. Follow-up GCN-based work mainly contains adaptive spatial graph kernels or bi-directional skeleton sequences [35,36,37,38], and $C_{k}$ kernel in paper [35] calculates the similarity of any two joints in each frame and is similar to the attention mechanism. Qin and Liu [39] introduced angular encoding to fuse higher-order features in GNN at the channel dimension of original inputs. Although the Transformer architecture [40] with self-attention module [41] as its mainstream has become the de facto standard for natural language processing (NLP) tasks, its structure has been widely applied in various research fields containing computer vision [42] and long sequence time-series forecasting [43]. In the field of skeleton-based human action recognition, researchers have proposed methods such as the spatial–temporal Transformer network (ST-TR) [7] and the group activity recognition network GroupFormer [44]. These methods utilize Transformer self-attention operators to model dependencies between joints and capture spatial–temporal contextual information jointly. It has been observed that Transformer-based methods have achieved superior performance compared to GCN-based methods using skeleton coordinates as inputs.

Video understanding is another human action recognition field using video clips as inputs [45]. Clips in these video datasets are usually taken from Internet videos covering a broad range of human action-related classes. Karpathy et al. [46] collected the Sports-1M dataset and proposed a multi-resolution CNN architecture with a context stream that models low-resolution images and a fovea stream that processes high-resolution center crops. Simonyan and Zisserman [47] extracted optical flow and proposed a two-stream ConvNet to capture spatial and temporal information using a single frame image and multi-frame optical flow. Jing et al. [48] tried to recognize actions in complicated scenes. Specifically, they presented a spatio-temporal neural network model with a joint loss to recognize human actions in videos, and they used a two-stream network extracting optical flow and appearance to capture spatial features and a LSTM network to model temporal dynamics. Tran et al. [49] extended 3×3 2D to 3×3×3 3D convolution kernels, and learned C3D features aligned by a simple linear classifier were used for video understanding. Three-dimensional CNNs are hard to train due to the scarcity of videos. Carreira and Zisserman [4] inflated 2D kernel and proposed a two-stream inflated 3D ConvNet (I3D). The I3D model can be initialized with corresponding 2D CNN net pre-trained on a large image dataset that helps the network training. As extracting optical flow from the videos is time-consuming, some work [50,51] has tried to apply hallucination-based methods learning to augment the network inputs. For example, Wang et al. [50] tried to transform the output of an I3D feature map into the Fisher vector representation via a hallucination step. Tang et al. [51] proposed the network to imagine the optical flow features from the appearance inputs for saving computation costs. Feichtenhofer et al. [52] presented a SlowFast network framework, in which the slow pathway has a low frame rate and higher channels to capture spatial features, and the fast pathway can learn temporal embeddings with a high frame rate and lighter channels. Three-dimensional CNN kernels such as C3D and I3D can be filled into the framework for recognition tasks. With the popularity of the attention mechanism, work [53] applied the spacetime non-local operation to the task of video recognition, and the authors of paper [5,54] adapted the standard Transformer architecture to videos with different skills on large-scale video dataset recognition.

In our prior work [12], we utilized 3D skeleton coordinates as the input for Tai Chi action recognition. However, due to the differences in the data format and collection system between the pre-trained NTU RGB+D dataset and the Tai Chi dataset, the recognition performance was relatively poor even after applying batch normalization to the network pipelines. In order to address this issue, Duan et al. [13] proposed the use of 3D heatmap volumes instead of traditional graph sequences as a representation of human skeletons, which has several advantages over 3D skeleton coordinates. For instance, since measurements may differ across different datasets, using 3D skeleton coordinates for the same action may lead to significant differences that require normalization across datasets. The use of 3D heatmap volumes can directly resolve such normalization problems, in addition to allowing for the application of video processing tricks such as centering and cropping to enrich the training dataset. Moreover, multi-person action skeletons can be projected onto a single heatmap, and we can utilize mainstream methods in the video understanding area to achieve steady and better performance without additional computation costs. However, the computation of GCN scales linearly with the increasing number of samplers in one action. In this work, we aim to transform graph-based skeleton coordinates into 3D heatmap volumes to enable the sharing of human action features across different datasets with alleviated data format mismatches. For feature extraction, we will use the PoseConv3D network as our model backbone, which has demonstrated superior performance compared to prior action recognition methods.

### 2.3. Research on Small-Scale Fine-Grained Datasets

Researchers strive to improve recognition performance on popular large-scale coarse-grained datasets with novel neural networks. The establishment of specific small-scale fine-grained datasets could have been better. Weinland et al. [55] created an IXMAS dataset containing 11 actions from multi-cameras. Nicora et al. [56] established the MoCA dataset consisting of 20 fine-grained cooking actions from 3 camera views for each activity. Moreover, MoCA is a bi-modal dataset collecting motion capture data and video sequences in a cooking scenario. These specific datasets can testify to the fine-grained recognition ability of the proposed model, which can help the application in specific scenarios. Gu et al. [57] published a fine-grained basketball action dataset consisting of annotated basketball game videos. They proposed a two-stream network integrated with NTS-Net to extract discriminative features for their fine-grained dataset.

Wu and Shao [58] proposed a multi-max-margin support vector machine (MMM-SVM) to improve IXMAS dataset accuracy with a multi-view system. Wang et al. [59] used an internal transfer learning strategy to enhance small-scale dataset performance. This strategy did not utilize other datasets, and the best model needed to be selected from the candidate model list, which needed to be more complex for application. Shen et al. [60] presented an automatic data augmentation model called Imaginative Generative Adversarial Network that can sample new data from learned dataset distribution. Augmented datasets can improve the classification accuracy with the same neural network. This method needs relatively large-scale datasets, and small-scale datasets were still hard to train because of a lack of samples. Ijaz et al. [6] presented a multi-modal transformer-based network to extract and fuse feature information from skeletal joints and acceleration data for improving small-scale fine-grained nursing activity [61] recognition performance. However, they did not pre-train the network with the NTU-RGB+D dataset and only introduced adaptive sharpness aware minimization (ASAM) [8] to converge their transformer models. They also inferred in their conclusions that one can also explore pre-training the skeletal branch to further improve the model’s convergence, which has been implemented in our work, and stated the superiority of the pre-trained pipeline. Goyal et al. [62] tried to take advantage of large-scale pre-trained representations under the hypothesis that they implicitly incorporate relevant cues for the small-scale dataset task. That paper also applied view-wise batch normalization to minimize the internal covariate shift for the cross-view action recognition task. This method gave an experimental analysis of which layer output can be extracted as motion features and did not give a detailed description of implicit relevant cues among human datasets, all of which will be solved in our paper.

Unlike prevailing coarse-grained actions, Tai Chi action can be treated as a professional action category, and each Tai Chi action consists of multiple meta movements coordinated with body movements. Compared with daily human actions, attention needs to be paid to the movement characteristics of different human body parts at various stages. Recognizing Tai Chi action is challenging for the similarity among Tai Chi action classes. As Tai Chi has become an Asian Games event, a sound recognition network can help abecedarians to evaluate their activities. In Tai Chi related action recognition work, Lin et al. [63] proposed a large depth-included human action (DHA) video dataset containing 17 categories. They treat all Tai Chi actions as the same class. Sun et al. [64] proposed a fine-grained Tai Chi dataset of 2772 samples of 58 Tai Chi actions. All video samples were collected from websites with dynamic backgrounds. They also applied improved dense trajectory features and Fisher vector representation for recognition and achieved 51.39% recognition accuracy. Dong et al. [65] also proposed a Tai Chi dataset called ‘Sub-Tai chi’ consisting of 15 actions and applied structural LSTM with an attention module for recognition; they reached 79% recognition accuracy on their own dataset. Liu et al. [66] applied the ST-GCN model on their own Tai Chi dataset and achieved 89.22% recognition accuracy. All these studies created their own Tai Chi datasets and are not available on websites for comparison. They proposed specific or applied a popular deep learning method for recognition that is not general for fine-grained action recognition.

Our previous work conducted preliminary research on Tai Chi actions. We first created our Tai Chi action dataset and proposed a Tai Chi action recognition algorithm using node trajectory features [19]. We extracted hand-crafted features, and only a single node was used for feature extraction, so it did not perform well under small-scale training sets and could have had better generalization. We introduced deep learning methods and proposed a spatial Transformer network for Tai Chi action recognition [12]. We used the NTU RGB+D dataset to pre-train the model to overcome the over-fitting network problem. Tai Chi training samples were only responsible for the final classification head training with frozen model backbone parameters. This algorithm used 24 joints of the human skeleton and achieved improvement compared with the first conventional method. However, it only partially utilized 72 skeleton joints of the Tai Chi dataset, leading to missing information for our fine-grained Tai Chi action. Complex data pre-processing was also needed for network recognition. In addition, the performance under small training sets still needed improvement, and few analyses of the results were shown to readers. Our paper containing accuracy improvement and ablation studies will solve these weaknesses.

We intend to provide a general framework for these small-scale fine-grained action datasets in our work. We suppose that samples share common motion features among the human action datasets and propose a simple but effective model framework to solve the small-scale fine-grained action recognition problems, which have been evaluated on the public Nursing Activity dataset and our self-built Tai Chi dataset. The model framework does not bring adversarial networks and can recognize the small-scale fine-grained action dataset.

### 2.4. Pre-Training Strategy and Evaluation Protocol

In the computer vision and video understanding area, many researchers verify their methods by *linear evaluation protocol* or *fine-tune evaluation*. In linear evaluation protocol, we freeze the pre-trained network parameters and only train the final classifier with the target dataset. Furthermore, all network parameters are fine-tuned with the new dataset in fine-tune evaluation. For example, MoCo-V1 [67] first performed unsupervised pre-training on dataset ImageNet-1M [9], then froze the parameters and re-trained a supervised linear classifier. The main goal of this unsupervised learning is to learn transferable features. This strategy can testify to the effectiveness of extracted features. Simonyan and Zisserman [47] measured the performance of their spatial stream ConvNet using three evaluations containing training from scratch on the target UCF-101 dataset [3], linear evaluation protocol of pre-training on the ILSVRC-2012 dataset, and training the classifier on UCF-101, and fine-tuning on UCF-101. The I3D model [4] proposed inflated 3D CNN kernels, which can be pre-trained on 2D ImageNet models and bootstrap relative 2D CNN kernels to 3D CNN kernels for initialization. The I3D model also evaluates superior transfer learning ability by first pre-training Kinetics and fine-tuning on HMDB-51 [68] and UCF-101 datasets.

As noted above, the pre-training strategy is always paired with evaluation protocols. In downstream tasks of the computer vision field, such as object detection [69], semantic segmentation [70], and human pose estimation [71], researchers applied the network model pre-trained on the ImageNet dataset to fine-tune their downstream tasks. In the video understanding area, the I3D model used the parameters of the 2D convolution network pre-trained on the ImageNet dataset to initialize the weight parameters of its 3D convolution network, which can accelerate the convergence of the network. We observe that datasets in these downstream tasks are all natural images or video samples. Moreover, models pre-trained on the ImageNet hold a similar phenomenon that extracted features gradually varied from general in the first layer to task-specific in the last layer of the hierarchical network [72]. Specifically, the first few layers can capture low-level features of samples, such as color blob features of images, whereas outputs of the last layer are high-level and task-specific. Therefore, the model backbone pre-trained on ImageNet can extract low-level edge and texture features of natural images and accelerate the convergence of the model in downstream tasks of the other natural image or video datasets to achieve better performance compared with training from scratch.

In this paper, we intend to construct a general and accurate model framework using SlowFast as the model backbone. Unlike skeleton-based human action recognition methods, we transform skeleton-based action sequences into heatmap-based pseudo videos. We utilize large-scale datasets to extract common low-level motion features of the heatmap, which can help improve the recognition performance on our small-scale fine-grained action dataset. Former skeleton-based action recognition tasks just trained their proposed model from scratch using a large-scale skeleton-based action dataset NTU RGB+D. For a sufficiently large-scale dataset, the model trained from scratch can also show impressive performance [7,13]. However, in the downstream tasks of small-scale datasets, the pre-training strategy is an effective way to improve recognition performance [73]. Therefore, our model framework based on the pre-training strategy is necessary for small-scale action recognition tasks, which has never been addressed in previous work [6,57,59,60]. We also give a complete analysis with intuitional t-SNE [14] visualization to demonstrate the effectiveness of the pre-training strategy.

Our action research uses two evaluations for different purposes. Linear evaluation protocol demonstrates that our pre-trained model can learn common human kinematic features through heatmap-based pseudo video samples, and fine-tune evaluation experiment shows the transfer learning ability of our framework and superior action recognition performance compared with training the network from scratch.

## 3. Methods

We give a detailed description of our conversion of an action sequence from 3D skeleton joint coordinates to a 3D heatmap volume in Section 3.1. Then we introduce our model framework and training strategy in Section 3.2 and Section 3.3.

### 3.1. From Joint Coordinates to a 3D Heatmap Volume

For the PoseConv3D method, the action data should be transformed into a 3D heatmap volume. In the PoseConv3D method [13], they use 2D pose estimators HRNet [10] trained on COCO-keypoints [74] to extract 2D human poses from action video clips directly. Nursing Activities and Tai Chi action datasets contain only 3D skeleton coordinates within a frame, and the conversion to the 3D heatmap volume is needed. We do not adopt the suggestion in PoseConv3D that said that an action sample is dividing a 3D skeleton (x,y,z) into three 2D skeletons, respectively, using (x,y), (y,z), and (x,z). Aiming for a general model framework, some datasets may be directly collected by cameras, and 3D coordinates extraction is infeasible or noisy, affecting the pre-trained or fine-tuned model. As shown in Figure 2d, axis *x* and *y* stand for the width and height of the sample, and axis *z* is the distance from the camera to samplers. As we did not fix the sampler-to-camera distance, the data format (y,z) and (x,z) lack physical meanings, so we only use 2D coordinates (x,y) to form a 3D heatmap volume, and the *z* axis coordinates can be used for data augmentation by rotating along the *y* axis.

As described in paper [11], one action is captured from three fixed camera viewpoints simultaneously, and we can obtain multi-view samples for the same action. We can also access multi-view samples using a perspective transformation by rotating along the *y* axis, illustrated in Figure 3b. The action sequence of 3D skeleton coordinates $X\in R^{T\times V\times C}$, where T,V,C denote the number of frames, joints, and coordinates, contains three skeleton coordinates for *V* skeleton joints collected by the motion-capture system. To make the most of our collected 3D skeleton sequences, we perform rotation and shear operations for each small-scale training sample. As we have 3D coordinates, we can rotate the original skeleton along the *y* axis with a random angle α using Equation (Equation 1) or shear [75] the skeleton using Equation (2). In Equations (Equation 1) and (2), the subscript ori corresponds to the original skeletal joint data denoted by x,y,z. The subscript rot represents the 3D coordinates obtained after performing rotation augmentation. Additionally, the subscript shr indicates the data acquired through shear augmentation; $a_{12},a_{13},a_{21},a_{23},a_{31},a_{32}$ are shear factors randomly sampled from [−β,β], and β is the shear amplitude. A processing sample is illustrated in Figure 3. We use shear and rotation augmentation methods to enrich our training samples.
(1)$$\left(\begin{array}{c} x_{rot}\\ y_{rot}\\ z_{rot} \end{array}\right)=\left(\begin{array}{ccc} cos\alpha & 0 & sin\alpha \\ 0 & 1 & 0\\ -sin\alpha & 0 & cos\alpha \end{array}\right)\cdot \left(\begin{array}{c} x_{ori}\\ y_{ori}\\ z_{ori} \end{array}\right)$$
(2)$$\left(\begin{array}{c} x_{shr}\\ y_{shr}\\ z_{shr} \end{array}\right)=\left(\begin{array}{ccc} 1 & a_{12} & a_{13}\\ a_{21} & 1 & a_{23}\\ a_{31} & a_{32} & 1 \end{array}\right)\cdot \left(\begin{array}{c} x_{ori}\\ y_{ori}\\ z_{ori} \end{array}\right)$$

For joint coordinates in every frame, we first perform an axis alignment because of different coordinate systems: $x_{cv}=-x_{pn}$, $y_{cv}=-y_{pn}$, in which subscript cv and pn stand for cv2 and perception neuron system, as shown in Figure 4a,b. Then the aligned data are subjected to a series of normalization operations containing minimize, scale-up, and centered, so the coordinates are normalized in a 1080×1920 resolution video canvas, the same as the NTU RGB+D dataset. The normalized 2D coordinates are transformed to a 2D keypoint or limb heatmap using Equations (Equation 3) and (4).

For every 2D joint coordinate $\left(x_{k},y_{k}\right)$, in Equation (Equation 3), ${\mathrm{Hk}}_{ij}$ is the keypoint heatmap value at pixel (i,j), $c_{k}$ is the confidence score of $\left(x_{k},y_{k}\right)$, and σ controls the variance of Gaussian maps; ${\mathrm{Hv}}_{ij}$ is the value of pixel (i,j) in the limb heatmap. As shown in Figure 5, intersections of the grid lines are pixels of the heatmap image, and *a* and *b* are two joints of a bone. We calculate the heatmap values of all these pixels using Equation (4). The expression d2_ab is the projection distance from the pixel to the bone $\vec{ab}$, so the value will always be 1 when the pixel is on the limb. After calculating heatmap values of all joints and bones, the final heatmap value of pixel (i,j) is the max value of these *V* heatmap values, illustrated in Equation (5). Figure 4 illustrates original 3D joint coordinates and a Tai Chi action sample’s final 2D keypoint and limb heatmap at one frame.
(3)$$\begin{array}{cc} {\mathrm{Hk}}_{ij}^{k} & =\exp^{-\frac{{\left(i-x_{k}\right)}^{2}+{\left(j-y_{k}\right)}^{2}}{2\sigma^{2}}}\times c_{k} \end{array}$$
(4)$$\begin{array}{cc} {\mathrm{Hv}}_{ij}^{k} & =\exp^{-\frac{d2\_{ab}^{2}}{2\sigma^{2}}} \end{array}$$
(5)$$\begin{array}{cc} H_{i,j} & =\max\{H_{i,j}^{1},\cdots ,H_{i,j}^{V}\} \end{array}$$

After finishing the 2D heatmap construction of 2D skeleton coordinates within a frame, we group them into a 3D heatmap volume across all frames. We need to reconstruct the heatmap channel because two datasets with different numbers of joints are used as inputs to the network. In this paper, we design two strategies to couple the information of the two datasets. One is to select the corresponding 17 skeleton joints from 72 Tai Chi joints to match the NTU RGB+D dataset. As shown in Figure 6, COCO-kp17 has more facial features but fewer hand features. Our perception neuron has only one head output joint, so we must approximate other facial features with the same joint. In addition, Tai Chi actions have rich hand expressions, and our 72-joint configuration can express hand gestures such as hook, fist, and palm. The second strategy is to split the whole skeleton into five parts—four limbs and a trunk—illustrated in Figure 6b,d. This method can keep the physical structure of human action and share all joint information of the two datasets.

Concatenating all 2D heatmaps along a temporal dimension, we construct a 3D heatmap volume $H\in R^{C\times T\times H\times W}$. In practice, we conduct a series of data processing strategies for the heatmap volume. We fix the volume to 48 by uniform sampling. We also follow the centered cropping strategy [13] for finding the smallest box that encloses all the 2D poses across frames, subsequent to a random crop, which is a popular processing method in video understanding. Heatmap volumes will resize to the resolution of 56×56 and have a 50% possibility to flip the right and left part of the skeleton to enrich datasets. Compared with other recognition methods, our pre-processing data method is more concise without zero-padding [34] or data interpolation strategy [12] and has no limitation on the number of samplers.

### 3.2. Model Backbone

Our 3D heatmap volume can be treated as a special video format. We take the 3D heatmap volume after joint segment process as the network input, illustrated in Figure 1. Our network backbone, shown in the middle model backbone part of Figure 1, follows the configurations of PoseConv3D [13], details of which are shown in Table 1. They utilized the SlowOnly [52] approach, which directly extends the ResNet layers from 2D to 3D in the last two stages as their model backbone.The middle column lists the components of each residual block, and the right illustrates output feature size after the corresponding block. Dimensions of kernels are denoted by $\left\{T\times S^{2},C\right\}$ for temporal, spatial, and channel sizes. Non-degenerate temporal filters (temporal kernel size > 1) are underlined. Residual blocks are shown by brackets, and the instantiated backbone is ResNet-50 [76]. We can see from Figure 1 that no downsampling operations are applied at the temporal dimension. Finally, action embeddings will pass through the linear classification head, containing a global average pooling (GAP) layer and a fully connected (FC) layer to generate a predicted action class.

Compared to the ResNet-50 model, we have removed the ResNet2 block from our model backbone, as we have already extracted the pose features from the videos using skeleton coordinates. We have elaborated in Section 2.4 that our hierarchical network gradually extracts features from general to task-specific. Our heatmap-based pseudo video samples do not contain any low-level features due to the lack of background information. Because our inputs are already mid-level features relative to the natural images, the network model with a shallow layer and a narrow channel is more suitable. Compared with X3D [77] and I3D models [4], our model backbone is lighter, which improves the feature extraction efficiency with heatmap-based pseudo video inputs. We intend to take denser temporal inputs with 48 uniformly sampled frames in each input sample, and the denser inputs enable us to capture more temporal dynamics for action recognition. There are no downsampling operations along the temporal dimension across the model framework, and the other operations follow the ResNet-50 structure to improve the model backbone’s robustness.

### 3.3. Training Strategy

In NTU RGB+D action recognition research, Duan et al. [13] achieved state-of-the-art accuracy in skeleton-based cross-view (CV) and cross-subject (CS) settings. However, they used 2D skeleton coordinates data extracted from videos and discarded 3D skeleton coordinates provided by the original dataset, which were used in other recognition methods. Figure 7a shows the visualization of original 3D skeleton coordinates and Figure 7b is the heatmap of extracted 2D coordinates from RGB sample video. For this action of drinking water, there are two people in the sampling environment and only one person is drinking water along with another interference sampler. Original 3D skeleton coordinates have two samplers, and the data are too blurry to recognize intuitively. Previous work filtered the original data and utilized denoised data for action recognition. This could extract the correct sampler from samplers, but the sampling environment always affects the data equality. When they failed to extract the correct sampler from a raw sample, they had to input both sampler data into the network and calculate the average score of extracted features. The NTU RGB+D datasets have interaction samples, so the number of samplers is varied according to the action class. Previous work had to pad the second person of a single-person action sample to a zero value, which is convenient for batch calculation. Figure 7b reflects a more accurate drinking water action and the video format has no zero-padding problem, which is concise for application. In our pre-training period, we continue to use extracted 2D coordinates from the NTU RGB+D video dataset, which helps us better learn common human action features and reduces the over-fitting of the network.

During the training process, illustrated in Figure 1, we first pre-train the model backbone with the NTU RGB+D dataset. For small-scale fine-grained action recognition, we separately experiment with linear evaluation protocol and fine-tune evaluation for freezing all pre-trained backbone parameters or fine-tuning the whole network to testify to the effectiveness of the proposed model framework. As we can generate 3D heatmap volumes for the joint and limb format, we also experiment with pseudo heatmaps and their fusion strategy, details of which are discussed in Section 4.

## 4. Experiments

The effectiveness of the pre-trained model is demonstrated by experimenting with Tai Chi action recognition, using both linear evaluation protocol and fine-tune evaluation. The datasets used in the experiment are briefly noted in Section 4.1, and the experiment settings, including the grid search of learning rate and our pre-trained models, are described in Section 4.2. Section 4.3 explores the property data augmentation method used in training the samples. Section 4.4 confirms our proposal that the uniform joint segmentation and pre-training method can improve action recognition by sharing more common human action features. Our recognition results using linear evaluation protocol on the Tai Chi dataset are presented in Section 4.5. In Section 4.6, we demonstrate the effectiveness of the pre-training period by fine-tuning the pre-trained model. Finally, in Section 4.7 and Section 4.8, we display the recognition results from the fine-tune evaluation and compare them with other methods on the Nursing Activity and Tai Chi datasets.

### 4.1. Datasets

The NTU RGB+D is a large-scale human action dataset collected by Microsoft Kinect V2. For our experiment, we selected the NTU RGB+D 120 dataset containing daily, mutual, and health-related actions as our pre-trained dataset. Dataset samples are extracted from corresponding video clips and hold the data format of COCO-kp17, shown in Figure 6a. We follow the cross-subject evaluation (X-Sub) criteria to ensure the model performance in the pre-training stage. In our experiment, X-Sub criteria are composed of 63,026 training and 50,919 testing samples. Our previous work [12] used the first 49 single-person classes of the NTU RGB+D 60 dataset for pre-training. However, in this work, we aim to provide a general and accurate transfer-learning framework for small-scale fine-grained action recognition. To achieve this, We use heatmap-based pseudo videos as network inputs, which have no limitations on the number and samplers. Due to the universality of 3D heatmap volume, we use all samples in the NTU RGB+D 120 dataset to construct a large-scale pre-training dataset, simplifying the pre-processing process of samples before feature extraction. A large-scale pre-training dataset can also enlarge the feature distribution of human actions, improving the ability of motion feature extraction of the pre-training model backbone, which has been proved in related work [73,78].

The public Nursing Activity dataset was collected at the Smart Life Care Society Creation Unit based on the collaboration between Kyutech and Carecom [61]. Similar to our objectives, they aim to promote research on specific work-related activities. They collected a multimodal dataset comprising six nursing care activities, which have been divided into 282 training and 116 testing samples. Each sample includes skeletal data and acceleration data, but we only utilized the skeletal data for small-scale nursing activity recognition to ensure a fair comparison. The categories of the six activities are presented below:Vital signs measurements;Blood collection;Blood glucose measurement;Indwelling drip retention and connection;Oral care;Diaper exchange and cleaning of area.

Our small-scale Tai Chi dataset has 10 Tai Chi actions, and each action has 20 samples, so the dataset contains 200 pieces. Details of our Tai Chi action dataset are available at website https://hit605.org/projects/taichi-data/ (accessed on 10 May 2023). Ten types of Tai Chi action names are as follows. For the sake of concise representation of the image, the category labels in the t-SNE feature maps are only marked with numbers.

Preparation;Grasp Bird’s Tail;Single Whip;Lift up Hand;White Crane Spread Its Wings;Brush Knee and Twist Step;Hold the Lute;Pulling, Blocking and Pounding;Apparent Close Up;Cross Hands.

The representation of each action’s name and iconic action is illustrated in Figure 8. A Tai Chi action is composed of a series of meta actions, and it is not easy to recognize with only one picture. For example, in Figure 8, actions ‘Brush Knee and Twist Step’, ‘Hold the Lute’, ‘Pulling, Blocking and Pounding’, and ‘Apparent Close Up’ have similar human poses with slightly different hand gestures. We have shown these four actions in Figure 9, and it is not easy to intuitively distinguish these fine-grained actions at our first glimpse. This requires the network to capture spatial embeddings and temporal dynamics. In addition, recognizing our small-scale Tai Chi dataset is very challenging for facing the over-fitting problem of the network.

### 4.2. Experiment Settings

For the pre-training period, we use SGD as our optimizer with momentum set to 0.9 and weight decay set to $3\times 10^{-4}$. We also use distributed the data-parallel framework (DDP) to accelerate training, as four GPUs are simultaneously used for calculation with a batch size of 32. We train for 24 epochs on the NTU RGB+D dataset, with the training dataset repeated ten times in each epoch at an initial learning rate of 0.1, and we use the cosine annealing schedule [79] to adjust the learning rate.

While finishing the pre-training on the NTU RGB+D dataset, we evaluate the Tai Chi dataset using linear protocol or fine-tune evaluation. We split 10%, 30%, 50%, and 70% of Tai Chi action samples as training samples, respectively, for action recognition. We repeat the training dataset 30 times and train 24 epochs on all configurations. Therefore, only 20 Tai Chi samples are collected as the training dataset in the ‘10% training’ configuration, which will increase the difficulty of network recognition. We only evaluate the Nursing Activity dataset using fine-tune evaluation and compare our results with previous SOTA methods to illustrate the effectiveness of our model framework.

We demonstrate that a pre-trained model can capture common human features, and we verify by linear evaluation protocol. In MoCo-V1 [67], they perform unsupervised pre-training on the ImageNet-1M dataset and test the accuracy of the ImageNet validation dataset. They perform a grid search and find the optimal initial learning rate is 30. Our experiment also performs a grid search to find the best initial learning rate in the linear protocol or fine-tune evaluation. We experiment with a 5-input-channel model with keypoint heatmaps as network inputs on Tai Chi dataset. We take 10% of Tai Chi samples as the training dataset in the linear evaluation protocol and fine-tune evaluation. Experiment results are shown in Table 2. In the linear evaluation protocol, the best initial learning rate is 0.4. We analyze that a significant initial learning rate will cause recognition failure because differences in our Tai Chi actions are tiny. The classifier cannot separate these fine-grained samples. When the initial learning rate continues to decline, recognition performance drops to a large extent. The model only adjusts the parameter weights of the final FC layer with our small-scale Tai Chi training dataset, and this learning rate will cause the problem of local optima. In the fine-tune evaluation, the best learning rate is 0.01. Because we use our small-scale Tai Chi samples to fine-tune all parameter weights of the model, a small learning rate is necessary to avoid the severe over-fitting problem. To summarize, we choose the initial learning rate as 0.4 and 0.01 in linear and fine-tune experiments for higher accuracy and faster training time.

As for the inference period in video understanding, a two-stream network [35] selects one frame from a video clip to capture spatial embeddings in the spatial stream. They uniformly sample 25 frames and obtain 10 inputs for each frame by cropping and flipping four corners and the center of that frame. The final recognition result of this video is the average score of these 250 frames and crops. In SlowFast network [52], they uniformly sample 10 clips from a video sample along the temporal axis and take three crops of 256×256 to cover the spatial dimensions. They also use the average softmax score for final prediction. In our inference period, we sample 10 clips from a test sample and take the average score as the last predicted action, similar to the above two methods.

We give a brief introduction to our experiment settings. Overall experiments can be divided into linear evaluation protocol and fine-tune evaluation. In the linear evaluation protocol, we testify the effectiveness of our joint segmentation method and the improvement of fusing results with comparison experiments. We also state that a pre-trained network can learn common human motion features that make the recognition easier for small-scale Tai Chi action recognition. We experiment using 5-channel or 17-channel keypoint or limb heatmaps. We also try to use early fusion or late fusion to find the proper fusing method. In fine-tune evaluation, we show that our model framework can reach high and steady performance under very few samples, and we experiment under different data pre-processing methods. We apply t-SNE [14] to show the embedding distribution of backbone outputs, which is simple and straightforward for our explanation.

In our model framework, the pre-training process is essential for extracting common human motion features. To demonstrate the effectiveness of our proposed model elements, we pre-train our model backbone under various settings using the NTU RGB+D 120 dataset with X-sub criteria. Recognition performance is illustrated in Table 3. We prepare different network inputs and name them for the simple description below. For example, “5-channel-kp”means taking our segmented 5-part keypoint format sample for pre-training and “10-channel-kplb” means that we concatenate the original 5-part keypoint and limb inputs along the channel dimension and take fused 10-channel samples as network inputs. The results show that all configurations have similar performance at about 85% recognition accuracy, giving us a comparable baseline for our fine-grained action recognition experiment. Detailed recognition performance and training logs can be found in our code repository at https://github.com/eunseo-v/SSCLS/tree/master/model_pth (accessed on 10 May 2023).

### 4.3. Effect of Data Augmentation in Linear Evaluation Protocol

In linear evaluation protocol, we freeze the weights of the backbone parameters during Tai Chi training and only train the final classifier. Proper data augmentation can enhance the features extracted from the frozen model backbone, thereby improving the classifier’s recognition accuracy. Our data augmentation strategy contains random flipping of the left and right joints of the human skeleton and random cropping of video samples. Experiments are under 10% training dataset configurations, and results are illustrated in Table 4. In Table 4, *WFWC* stands for “with flipping and with cropping”; *NFNC* refers to “no flipping and no cropping”.

The experimental results demonstrate that the combination of flipping and cropping operations enhances the recognition performance in the linear evaluation protocol. Specifically, we have frozen the model backbone parameters, and the effect of these operations is solely limited to the convergence of the final classifier during training. The flipping operation mirrors the original human body and is particularly useful for skeleton-based samples. On the other hand, cropping represents the variation in the position of the human body in the video. Both operations help maintain the human posture and augment the training dataset. Consequently, our subsequent experiments employ the *WFWC* configuration.

### 4.4. Effect of Joint Segmentation and Pre-Training Period in Linear Evaluation Protocol

The various datasets containing human action samples are obtained from multiple systems and feature different body joints. Our model pipeline cannot fully match the corresponding body joints among datasets, resulting in the loss of valuable data information. Therefore, we must re-design the network input for different datasets. For instance, the pre-trained NTU RGB+D dataset comprises 17 skeleton joints, whereas our Tai Chi action dataset includes 72 joints. As illustrated in Figure 4, NTU RGB+D offers more facial information, whereas the Tai Chi dataset contains more hand features. To match corresponding joints between the two datasets, we choose to use the same head joint in the Tai Chi dataset to approximate five facial joints in the NTU RGB+D dataset, and we discard expressive hand joints in the Tai Chi dataset in order to match the only corresponding hand joint in the NTU RGB+D dataset. However, this approach may result in the loss of crucial information when recognizing more fine-grained action datasets. To retain as much actionable information as possible when matching corresponding body joints, we divide the skeleton joints into five parts: four limbs and a trunk. Each part serves as a network input channel, enabling our model framework to extract common motion features using a pre-trained dataset and aid recognition under our small-scale fine-grained dataset.

We experiment under these two configurations: 17 joint matching and 5 part segmentation, using “17-channel-kp” and “5-channel-kp” as pre-trained models, respectively. We use 90% of total Tai Chi samples as the testing dataset in the linear protocol evaluation. We apply t-SNE [14] visualization to the features after the model backbone and show feature distribution in Figure 10. Feature embeddings are visualized with different colors, and we can judge the feature extraction performance of the model backbone from t-SNE visualization.

In Figure 10, we observe that Figure 10a has denser feature distribution and fewer isolated points than Figure 10b. This indicates that our joint segmentation strategy can extract common motion features and reach 5% higher recognition accuracy than the joint matching strategy.

To show the pre-training process can learn common features and is helpful for fine-grained Tai Chi action recognition, we also initialize the model backbone and freeze these parameters to evaluate the classifier performance. Results are shown in Figure 11. Although purple points in Figure 11b gather together, which indicates that this action has discriminative motion features, most samples are blurry and hard to distinguish. After training on the Tai Chi dataset, the model with random init backbone parameters reaches 10% recognition accuracy, which fails to be recognized. These two experiments have shown the joint segmentation strategy’s superiority and the necessity of the pre-training process.

### 4.5. Recognition Results in Linear Evaluation Protocol

We conduct an evaluation of our Tai Chi action recognition task using linear evaluation protocol under different training dataset settings. Specifically, we take 90%, 70%, 50%, and 30% of Tai Chi samples as testing datasets and utilize the same *5-channel-kp* pre-trained model backbone for evaluation. The resulting performance metrics are shown in Table 5. The findings from Table 5 show that a pre-trained model can extract common motion features that aid in Tai Chi recognition tasks across all configurations. Notably, there is a consistent trend of recognition improvement as training sample size increases, indicating that a rich dataset can provide abundant motion features that the classifier can leverage to produce credible recognition results. We also observe that the recognition accuracy cannot reach 100% even when using 70% of samples as the training dataset. This finding can be attributed to the fact that the parameters of the model backbone are frozen, and, therefore, the fine-grained Tai Chi dataset cannot be fully recognized through training the final classifier alone. This observation highlights the unique motion characteristics of the Tai Chi dataset that distinguish it from the pre-training NTU RGB+D dataset.

To verify the robustness and scalability of our proposed model framework, we develop two fusion methods for integrating extracted features from keypoint and limb format inputs. Specifically, we employ two fusion methods, concatenating two information types along the channel dimension (early fusion) or averaging the final features prior to the *softmax* function of two separate outputs (late fusion). Although previous research [80] attempted to fuse spatial and optical flow features at middle convolutional layers and identified the optimal fusion layer through experimentation, we pursued an alternative approach to identify the optimal fusion layer that aligns with our proposed general and scalable model framework. Unlike the lateral connection used in the SlowFast model [52] that requires separate training with two loss functions, using only one loss function with two streams can lead to severe overfitting. In contrast, the two-stream Conv Nets [47] employed early fusion and late fusion methods, enabling full utilization of the datasets’ information to achieve feature fusion for multi-modal datasets.

To identify the optimal fusion method for fine-grained action recognition, we conduct experiments under the “TEST9” configuration. For the early fusion method, we initially use “10-channel-kplb” as our pre-trained model, and in the late fusion experiment, we utilize respective “5-channel-kp” or “5-channel-lb” as the pre-trained model. Table 6 illustrates that the recognition performance using the late fusion method outperforms any simple keypoint or limb input and is significantly superior to the early fusion method. Our results indicate that a network using separate keypoint or limb format inputs can capture complementary motion features that aid in final fusion recognition, leading to improved small-scale fine-grained action recognition performance. Furthermore, this fusion method can be reasonably extended to multi-modality action recognition by taking the average feature of each single-modality output as the final result.

### 4.6. Effect of Pre-Training Period in Fine-Tune Evaluation

We propose that the pre-training strategy can help improve recognition performance on small-scale fine-grained datasets. To demonstrate this, we conduct two experiments under the “TEST9” configuration using our small-scale Tai Chi action dataset. We fine-tune the pre-trained model and also train the model directly from scratch, and we show the recognition performance in Figure 12 using the t-SNE feature distribution of model backbone outputs.

Figure 12a depicts the backbone output distribution with the pre-training process. We can observe that the features of the same action are closely clustered, and there is a clear boundary between any two actions. The pre-trained model achieves 94.44% accuracy in recognizing our Tai Chi dataset after the final FC layer. On the other hand, Figure 12b depicts the feature distribution without the pre-training process. The purple dot cluster and the red dot cluster are still gathered, which illustrate that corresponding actions, ‘Preparation’ and ‘Cross Hands’, are easier to distinguish. In Section 4.1, we demonstrated and analyzed that Tai Chi action categories 6 to 9 have similar human poses with slightly different hand gestures. Feature distribution in Figure 12b also confirms this statement. Features of these actions fall in the same place. That is to say, these samples are challenging in the recognition task. Because there is no boundary between different actions, the model can only reach 37.78% recognition accuracy on the same test samples. This experiment confirms the necessity of the pre-training strategy for small-scale human action recognition tasks, which we analyzed in Section 2.4. We demonstrate that features extracted from the pre-trained model are more discriminative and task-specific. Conversely, the model trained from scratch suffers the over-fitting problem because the network needs to extract low-level features within limited training samples, which impacts recognition.

### 4.7. Recognition Results in Fine-Tune Evaluation

In order to assess the stability of our model during fine-tuning, we conducted various experiments employing different data pre-processing configurations. Specifically, our experiments were conducted using the *TEST9* configuration. The recognition metrics are presented in Table 7, and Figure 13 displays the corresponding t-SNE visualization. Our results demonstrate that our proposed model framework is capable of achieving consistent and precise recognition performance, with extracted features that are more discriminative than those obtained from a model trained from scratch, as shown in Figure 12b. These findings attest to the effectiveness of our approach.

### 4.8. Comparison with Other Methods

In this paper, we present a comparison of our Tai Chi action recognition results with those of other methods, as summarized in Table 8. Xu et al. [19] employed a conventional SVM approach to recognize segmented trajectories, rather than deep learning techniques, for Tai Chi action recognition. Yuan et al. [12] proposed a general model framework for extracting discriminative motion features on small-scale, fine-grained datasets. In a paper using the spatial Transformer method [12], they observed notable improvements in recognition performance as the proportion of the training dataset increased. However, their approach still requires improvement in scenarios with limited training data. Compared with method [19], we entirely use Tai Chi action information and reach higher accuracy with a more straightforward network design.

In this paper, our method performs well even in the *TEST9* settings, which only reached 87.22% recognition accuracy in paper [12]. It means that only two samples in each Tai Chi action class are used for training, and it can reach 94.44% action recognition accuracy over the rest of the test samples with the pre-trained model. This confirms that our pre-training period can extract common motion features that help small-scale fine-grained Tai Chi action recognition and states the effectiveness of our proposed model framework.

We also evaluate our model framework on the publicly available Nursing Activity dataset and compare it with previous SOTA methods to state the superiority of our model framework. Previous studies by Cao et al. [81] applied ST-GCN directly to process 3D motion capture data. Ijaz et al. [6] presented a multi-modal transformer-based network (MMT) to extract and fuse feature information from skeletal joints and acceleration data for improving recognition performance. However, both studies used linear interpolation methods to complete the missing skeleton data, which may introduce noise information. Furthermore, they trained their models from scratch without any pre-training strategy. Our approach, which only employs skeleton modality samples and a pre-training strategy, outperforms these methods to a large extent, as demonstrated in Table 9. Notably, our model framework achieves superior results compared to MMT, which separately experimented with skeleton modality samples and fused acceleration modality together. Despite fusion results outperforming the skeleton model by 3%, both accuracies fall short of our performance. These findings demonstrate the generality and accuracy of our model framework. Finally, we present the confusion matrix of our experiment on the Nursing Activity dataset in Figure 14. Analyzing the matrices, we observe that our recognition accuracy in categories 1, 3, 4, and 5 is significantly higher than that of the MMT method, whereas we achieve similar recognition performance in categories 2 and 6. Considering that we only used skeleton samples and a more concise network structure, the improved recognition accuracy strongly demonstrates the effectiveness of our proposed model.

## 5. Conclusions

This paper presents a model framework designed for recognizing small-scale fine-grained actions. By leveraging anthropometric kinematics as prior knowledge, we can extract common human motion features from large-scale human action datasets. Our pre-trained model was evaluated using a linear evaluation protocol, which demonstrated its ability to capture common motion features. We employed the t-SNE method to visualize the extracted features. Furthermore, fine-tune evaluation experiments were conducted to demonstrate that our model framework can achieve consistent and precise recognition performance even with a limited number of training samples. We introduced a late fusion method for multi-modality action datasets to enhance recognition performance. Our model framework’s effectiveness was demonstrated through experiments conducted on the public Nursing Activity dataset and our self-built Tai Chi dataset.

In our method, the pre-training phase aims to acquire general human action features, but the recognition task on the NTU RGB+D dataset is considered over-sufficient rather than essential. Drawing inspiration from unsupervised research, our future work will focus on developing a more efficient and adaptable pretext task that facilitates the acquisition of general human features and enhances the performance of action recognition.

## Figures and Tables

**Figure 1 sensors-23-06364-f001:**
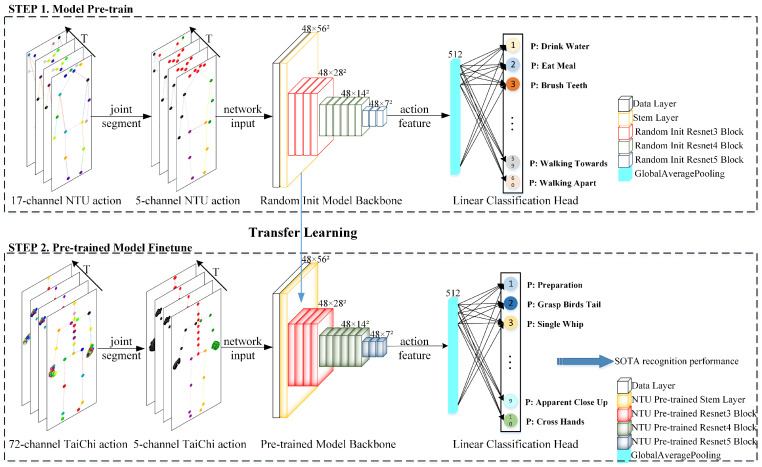
SSCLS model framework. The segmentation of skeleton joints into five parts, denoted by ’black’, ’green’, ’purple’, ’yellow’, and ’red’, encoding for four limbs and the trunk, respectively, is shown. A pre-training period was employed to train an action feature extraction network using large-scale datasets. During the fine-tuning period, the pre-trained model backbone, represented by padded-color cubes for distinction, was utilized for small-scale fine-grained action recognition tasks.

**Figure 2 sensors-23-06364-f002:**
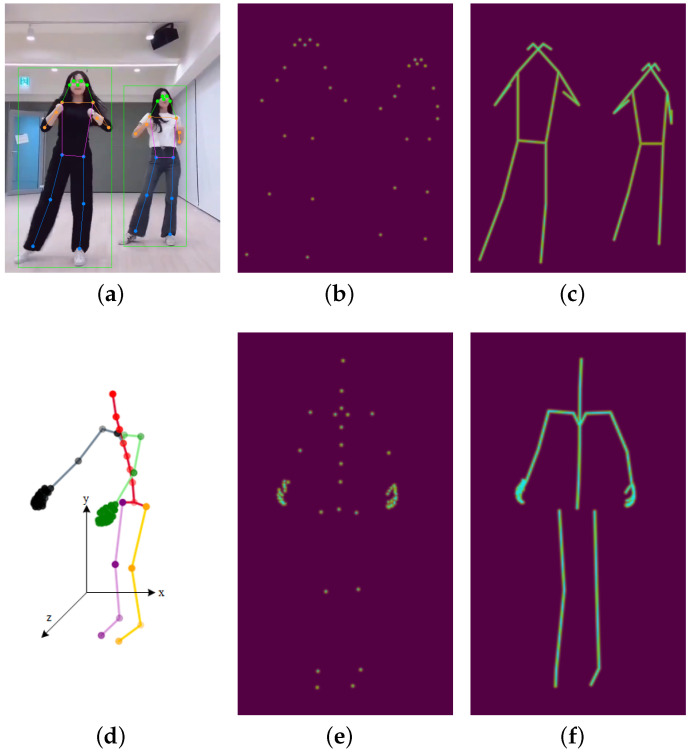
Heatmap visualization extracted from two modalities. For video images, we perform human detection and estimate 2D poses for each frame. For skeleton coordinates, we plot (x,y) coordinates on the heatmap canvas. (**a**) Estimated 2D poses; (**b**) 2D keypoint heatmap; (**c**) 2D limb heatmap; (**d**) original 3D coordinates; (**e**) 2D keypoint heatmap; (**f**) 2D limb heatmap.

**Figure 3 sensors-23-06364-f003:**
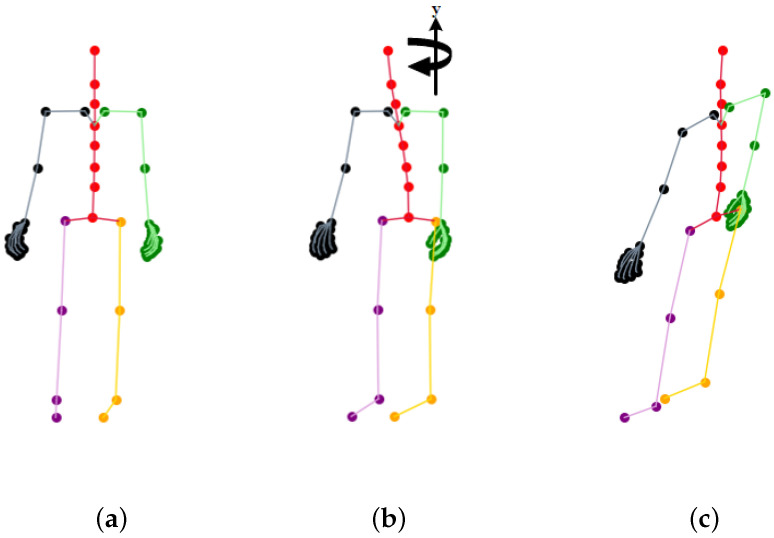
Skeleton image before and after rotation and shear operation. Rotated skeleton coordinates are transformed by rotating along the *y* axis, and the sheared skeleton can be regarded as projecting the skeleton into a new non-vertical platform. (**a**) Original skeleton; (**b**) rotated skeleton; (**c**) sheared skeleton.

**Figure 4 sensors-23-06364-f004:**
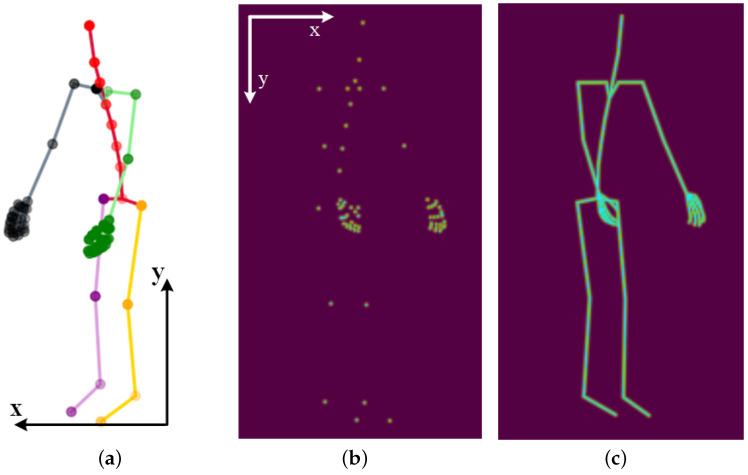
Three-dimensional joint coordinates and 2D heatmaps of Tai Chi action. We transform the sequence of the skeleton-based coordinates into a heatmap-based pseudo video, which is more suitable for our network model. (**a**) The 3D skeleton coordinates under the pn system; (**b**) 2D keypoint heatmap under the cv2 system; (**c**) 2D limb heatmap under the cv2 system.

**Figure 5 sensors-23-06364-f005:**
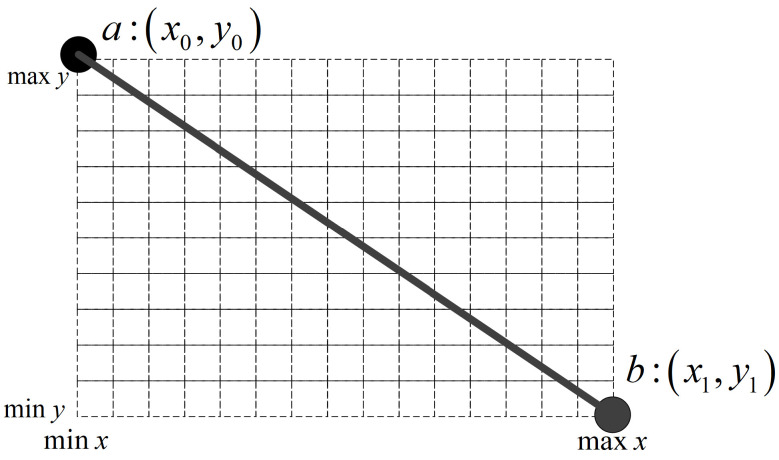
Illustration when calculating the heatmap of limb modality. For two joints of a bone *a* and *b*, we calculate the heatmap value of pixels at the intersection of all grid lines as shown in the figure.

**Figure 6 sensors-23-06364-f006:**
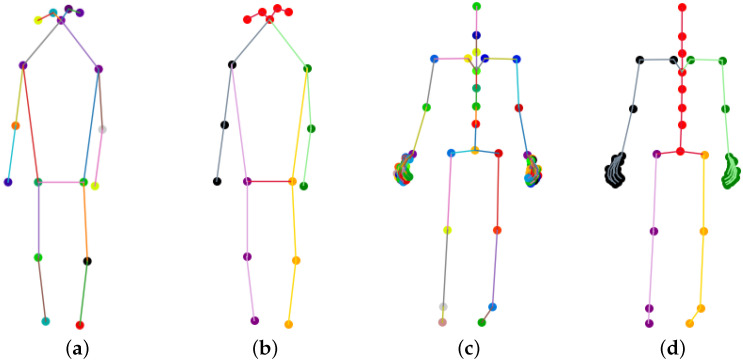
Skeleton joints comparison between datasets. COCO-17 contains more facial keypoints but lacks hand features. Our perception neuron system can capture abundant hand expressions, but only one head joint is included. Therefore, our joint segmentation method split the whole skeleton into five parts, shown in (**b**,**d**). (**a**) Seventeen channel COCO-17 skeleton coordinates; (**b**) 5 channel COCO-17 skeleton coordinates; (**c**) 72 channel perception neuron skeleton coordinates; (**d**) 5 channel perception neuron skeleton coordinates.

**Figure 7 sensors-23-06364-f007:**
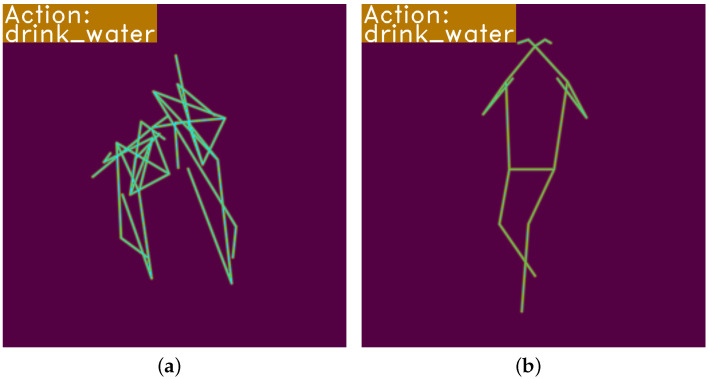
Comparison of original 3D skeleton coordinates and extracted 2D poses for sample ‘S012C002P017R002A001’. The heatmap in (**a**) is constructed by Kinect-collected 3D skeleton coordinates with method noted in Section 3.1, whereas the heatmap in (**b**) is directly extracted from videos with model HRNet [10]. (**a**) Heatmap from skeleton data; (**b**) heatmap from pose extraction.

**Figure 8 sensors-23-06364-f008:**
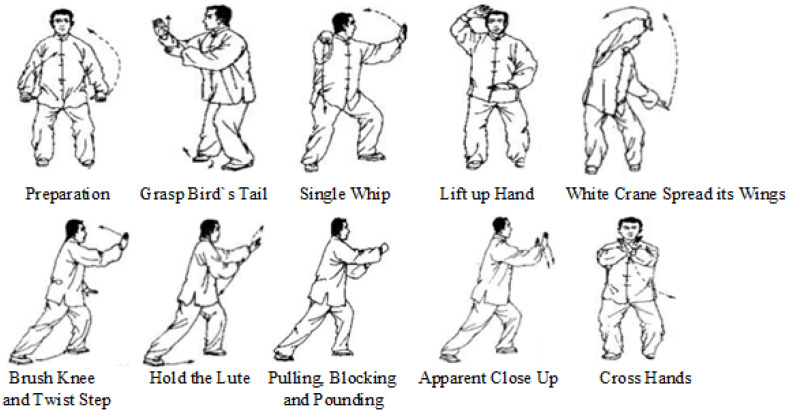
Illustration of name and the iconic frame of 10 Tai Chi actions. Actions among the Tai Chi dataset are similar, especially for actions ‘Brush Knee and Twist Step’, ‘Hold the Lute’, ‘Pulling, Blocking and Pounding’, and ‘Apparent Close Up’.

**Figure 9 sensors-23-06364-f009:**
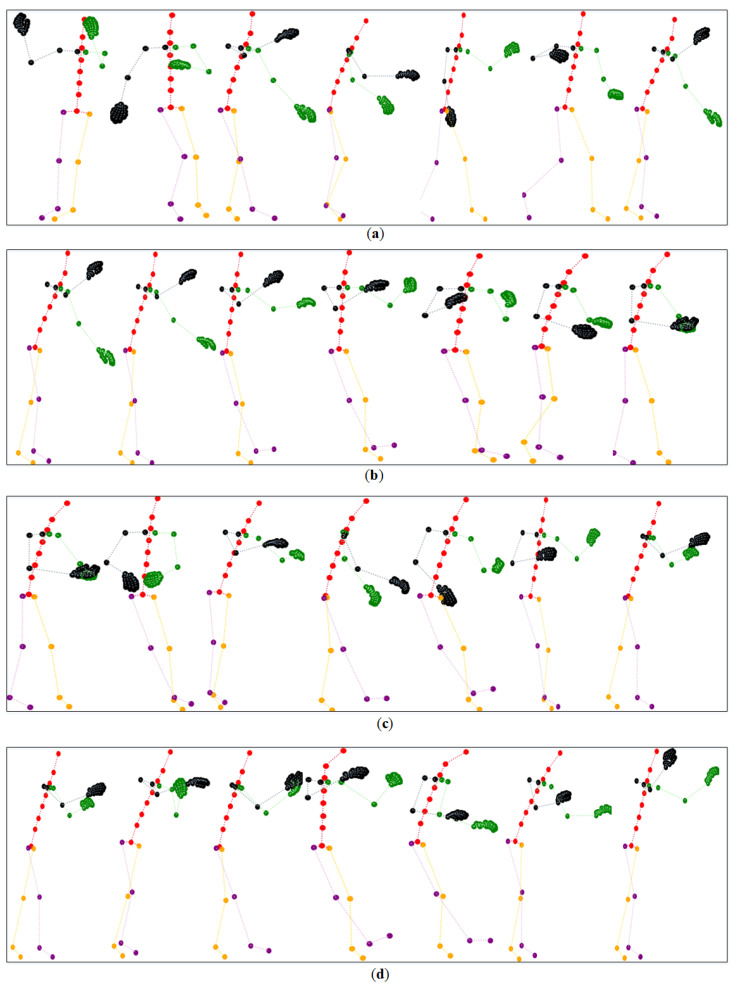
Skeleton sequence illustration of four similar Tai Chi actions. Unlike mainstream datasets, fine-grained Tai Chi actions bring more recognition difficulties for the proposed network. (**a**) Illustration of Action “Brush Knee and Twist Step”. (**b**) Illustration of Action “Hold the Lute”. (**c**) Illustration of Action “Pulling, Blocking and Pounding”. (**d**) Illustration of Action “Apparent Close Up”.

**Figure 10 sensors-23-06364-f010:**
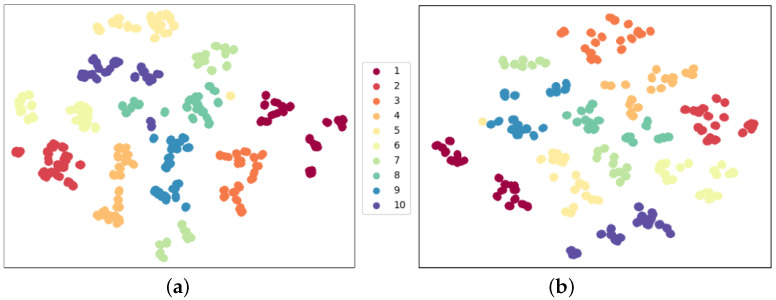
Feature distribution comparison using two joint matching strategies in linear evaluation protocol. The same color stands for the same action class. The specific action names corresponding to the category labels are listed in Section 4.1. (**a**) 5-channel-kp with acc 87.22%; (**b**) 17-channel-kp with acc 82.22%.

**Figure 11 sensors-23-06364-f011:**
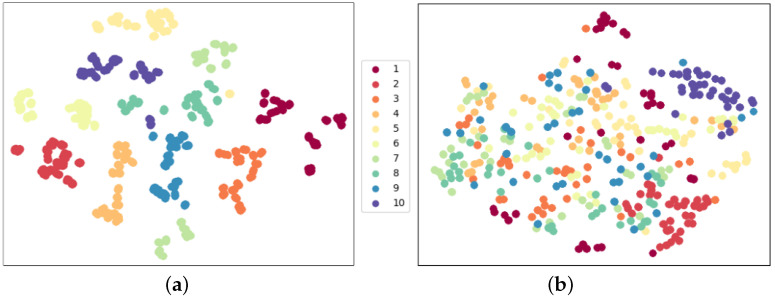
Feature distribution comparison with/without pre-training process. The same color stands for the same action class. The pre-trained model backbone can easily separate Tai Chi actions, whereas the random backbone collapses. (**a**) Pre-trained model backbone outputs; (**b**) random init model backbone outputs.

**Figure 12 sensors-23-06364-f012:**
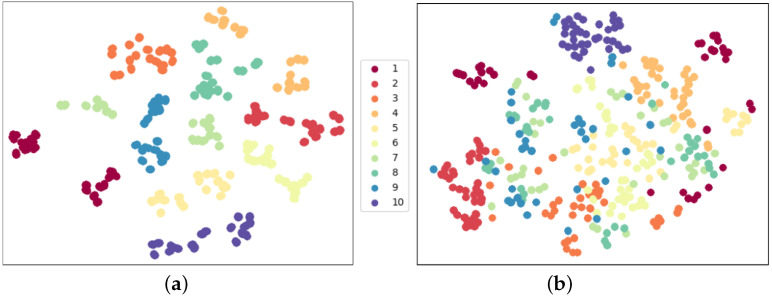
Feature distribution comparison with/without pre-training process in fine-tune evaluation. The non pre-trained model has an obviously severe over-fitting problem that t-SNE features with different actions stack in the same embedding space. (**a**) **WITH** pre-training process; (**b**) **WITHOUT** pre-training process.

**Figure 13 sensors-23-06364-f013:**
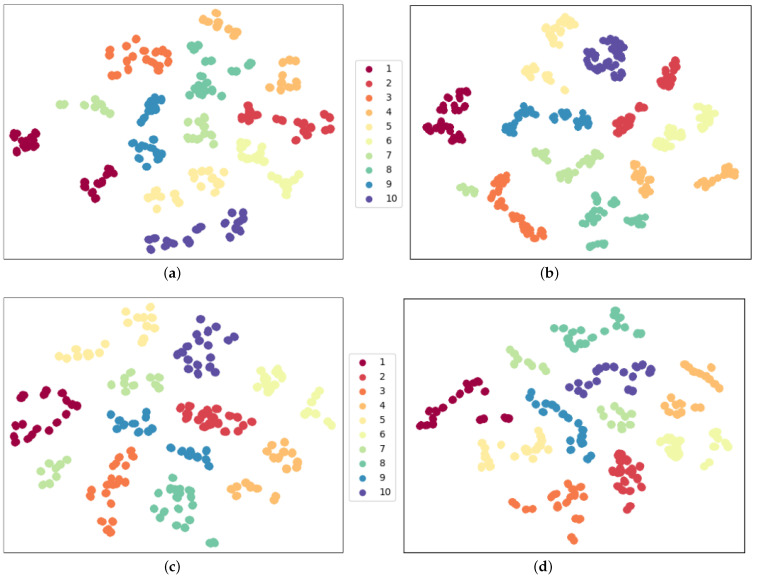
Feature distribution with different data pre-processing configurations in fine-tune evaluation. This supplements the t-SNE visualization of fine-tune recognition results in Table 7 and illustrates the robustness of our model framework. (**a**) WSWRNFNC accuracy = 94.44%; (**b**) WSWRNFWC accuracy = 94.44%; (**c**) WSWRWFNC accuracy = 96.67%; (**d**) WSWRWFWC accuracy = 98.33%.

**Figure 14 sensors-23-06364-f014:**
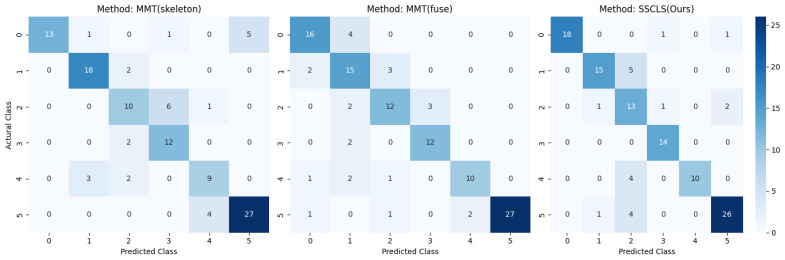
Confusion matrices of Nursing Activity dataset recognition using MMT and SSCLS methods. Activity index of the axis are 0: vital signs measurement, 1: blood collection, 2: blood glucose measurement, 3: indwelling drip retention and connection, 4: oral care, 5: diaper exchange and cleaning of area.

**Table 1 sensors-23-06364-t001:** Configurations of our model backbone. The dimensions of kernels are denoted by {$T\times S^{2},C$} for temporal, spatial, and channel sizes. Strides are denoted as {temporal stride, spatial stride^2^}.

Stage	Model Backbone	Output Size $T\times S^{2}$
Data layer	uniform $32,4^{2}$	$48\times 56^{2}$
Stem layer	conv $1\times 7^{2},32$ stride $1,1^{2}$	$48\times 56^{2}$
ResNet3	$\left[\begin{array}{c} 1\times 1^{2},32\\ 1\times 3^{2},32\\ 1\times 1^{2},128 \end{array}\right]\times 4$	$48\times 28^{2}$
ResNet4	$\left[\begin{array}{c} \underline{3\times 1^{2}},64\\ 1\times 3^{2},64\\ 1\times 1^{2},256 \end{array}\right]\times 6$	$48\times 14^{2}$
ResNet5	$\left[\begin{array}{c} \underline{3\times 1^{2}},128\\ 1\times 3^{2},128\\ 1\times 1^{2},512 \end{array}\right]\times 3$	$48\times 7^{2}$
GAP	Global Average Pooling	512
FC	Fully Connected Layer	Num Classes

**Table 2 sensors-23-06364-t002:** Grid search of initial learning rate under two evaluation configurations. We take the two bold initial learning rates in subsequent experiments for a fair comparison.

Initial Learning Rate	Accuracy (%)	Epoch
*Grid Search in Linear Evaluation Protocol:*		
30	82.78	23
3	84.44	23
1	83.89	16
**0.4**	**87.22**	**13**
0.1	85.00	18
0.01	78.89	7
*Grid Search in Fine-tune Evaluation:*		
1	52.22	23
0.1	87.22	24
**0.01**	**94.44**	**5**
0.001	93.89	11
0.0001	71.11	18

**Table 3 sensors-23-06364-t003:** NTU RGB+D 120 recognition results under ‘X-Sub’ criteria. e.g., ‘5-channel-kp’ stands for taking segmented 5-part keypoint format samples as network inputs.

Config	Accuracy (%)
5-channel-kp	85.12
5-channel-lb	85.00
17-channel-kp	85.42
10-channel-kplb	85.30

**Table 4 sensors-23-06364-t004:** Data augmentation recognition results in linear evaluation protocol. Abbreviations of ‘N’, ‘W’ means ‘not with’ and ‘with’; ‘F’, ‘C’ indicate operation of ‘flipping’ and‘cropping’, respectively. We use 10% of total samples as training dataset to train the classifier.

Config	Accuracy (%)	F1-Score (%)	Precision (%)	Recall (%)
NFNC	72.78	72.53	80.84	72.78
WFNC	73.33	72.87	78.33	73.33
NFWC	85.00	84.37	86.67	85.00
WFWC	87.22	85.54	89.74	87.22

**Table 5 sensors-23-06364-t005:** Fine-grained Tai Chi action recognition results in linear evaluation protocol. We experiment using keypoint modality from 10% training samples (‘TEST9’) to 70% samples (‘TEST3’) with fixed model backbone.

Config	Accuracy (%)	F1-Score (%)	Precision (%)	Recall (%)
TEST9	87.22	85.54	89.74	87.22
TEST7	96.43	96.40	96.90	96.43
TEST5	96.67	96.66	96.90	96.67
TEST3	98.00	97.99	98.18	98.00

**Table 6 sensors-23-06364-t006:** Fine-grained Tai Chi action recognition results across fusion methods in linear evaluation protocol. We experiment under ‘TEST9’ configuration with early fusion and late fusion methods. The ‘+’ indicates early fusion configuration, whereas ‘*√*’ indicates late fusion.

Config	Keypoint	Limb	Accuracy (%)	F1-Score (%)	Precision (%)
TEST9	*√*		87.22	85.54	89.74
TEST9		*√*	88.33	87.95	89.24
TEST9	+	+	85.00	82.58	87.80
TEST9	*√*	*√*	**90.56**	**89.82**	**91.55**

**Table 7 sensors-23-06364-t007:** Tai Chi action recognition results in fine-tune evaluation under *TEST9*. Abbreviations of ‘N’, ‘W’ mean ‘not with’ and ‘with’; ‘S’, ‘R’, ‘F’, and ‘C’ indicate operation of ‘shear’, ‘rotation’, ‘flipping’, and ‘cropping’, respectively. Our model framework achieves stable recognition performance under all data pre-processing configurations.

Config	Accuracy (%)	F1-Score (%)	Precision (%)	Recall (%)
WSWRNFNC	94.44	93.80	95.87	94.44
WSWRWFNC	94.44	93.66	95.48	94.44
WSWRNFWC	96.67	96.43	97.05	96.67
WSWRWFWC	98.33	98.29	98.47	98.33

**Table 8 sensors-23-06364-t008:** Comparison of Tai Chi action recognition accuracies using our Tai Chi dataset. We compare with SVM [19] and spatial Transformer [12] under four configurations: ‘TEST9’, ‘TEST7’, ‘TEST5’, and ‘TEST3’.

Config	SVM [19]	Spatial Transformer [12]	SSCLS (Ours)
TEST9	72.78%	87.22%	**94.44%**
TEST7	80.05%	90.71%	**100.00%**
TEST5	90.71%	93.00%	**100.00%**
TEST3	93.86%	98.33%	**100.00%**

**Table 9 sensors-23-06364-t009:** Recognition performance comparison with the SOTA methods. Our method outperforms these method in all evaluation metrics, which illustrates the superiority of our method.

Methods	Accuracy (%)	F1-Score (%)	Precision (%)	Recall (%)
ST-GCN [81]	56.59	56.61	56.55	59.41
MMT (skeleton) [6]	76.72	74.64	76.02	75.15
MMT (fuse) [6]	79.31	78.34	78.99	78.30
SSCLS (Ours)	**82.76**	**83.27**	**85.90**	**82.80**

## Data Availability

The dataset presented in this study is available. Raw data and pre-processing programs can be found on the website https://github.com/eunseo-v/SSCLS (accessed on 10 May 2023) under the section “Data Preparation”.

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
