# Peer review of "Improving Small-Scale Human Action Recognition Performance Using a 3D Heatmap Volume"

_sensors, 2023, doi:10.3390/s23146364_

Round 1
Reviewer 1 Report
This paper presents a model pipeline designed for recognizing small-scale fine-grained actions, which has certain theoretical significance and practical value. The detailed comments are as follows. Please consider them for further revision.
1、 On page 9, the parameters xrot, xori, xshr of Formula (1) and Formula (2) need to be introduced.
2、 On page 17, Table 2, for Grid Search in Linear Evaluation Protocol, please explain why 30, 3, 1, 0.4, 0.1, and 0.01 are selected as the initial learning rate instead of other values. In addition, for Grid Search in Fine-tune Evaluation, is it possible to use values such as 0.001, and 0.0001 for the initial learning rate? Please explain and add experiments if necessary.
3、 On page 23, Figure 14 only lists the confusion matrix of the proposed model SSCLS, and it is suggested to add the confusion matrix of the three methods ST-GCN, MMT(skeleton), and MMT(fuse) in Table 9 for comparison, which is more convincing for the paper.
4、 The English language of the manuscript should be improved.
This paper presents a model pipeline designed for recognizing small-scale fine-grained actions, which has certain theoretical significance and practical value. The detailed comments are as follows. Please consider them for further revision.
1、 On page 9, the parameters xrot, xori, xshr of Formula (1) and Formula (2) need to be introduced.
2、 On page 17, Table 2, for Grid Search in Linear Evaluation Protocol, please explain why 30, 3, 1, 0.4, 0.1, and 0.01 are selected as the initial learning rate instead of other values. In addition, for Grid Search in Fine-tune Evaluation, is it possible to use values such as 0.001, and 0.0001 for the initial learning rate? Please explain and add experiments if necessary.
3、 On page 23, Figure 14 only lists the confusion matrix of the proposed model SSCLS, and it is suggested to add the confusion matrix of the three methods ST-GCN, MMT(skeleton), and MMT(fuse) in Table 9 for comparison, which is more convincing for the paper.
4、 The English language of the manuscript should be improved.
Reviewer 2 Report
Major comments:
1. The authors state that the ResNet2 block have been removed from the feature extraction pipeline. Is that discussed? If yes, please supplement the reason, it is better if the author can give any experimental validation.
2. Is the purpose of this method to use it only to perform the classification of Tai chi datasets? If yes, please supplement the scenarios which the method is deployed for. If not. There are some literatures reported that the operation of pretrain-finetune method have a problem called "catastrophe forgetting". How to deal with this problem, please discuss it.
3. How does the author divide the dataset, which part of the dataset is trainset, which one is testset? Moreover, is the dataset shuffled? The authors should clearfy that.
4. The author proposes a space transformer network for Tai Chi action recognition, which should explain in detail how to overcome the overfitting problem of neural network in the use process. It is mentioned in the article that the overfitting phenomenon has been significantly alleviated and should be reflected.
Minor comments:
1. The citation in line 202 is missing.
2. The position of Fig.1 should be placed after you mentioned this figure in the content.
3. In line 461, the authors give a review on the literature [13]. This content should be placed in introduction section or discussion, so that the statement can be more concise.
4. SSCLS abbreviation not defined.
5. The quality of Figures 7 and 8 needs to be improved.
6. Tables must have a unified format.
Extensive editing of English language required
Reviewer 3 Report
Review:Improving Small-scale Human Action Recognition Performance Using a 3D Heatmap Volume
This paper presents a 3D heat map method to robustly identify human actions in videos via skeleton model analysis.
Are there joints that have a relevance to be present to correctly identify the action occurring?
The model or proposed model configuration os not clear, An figure or parts of figures must be included to support the model and text description.
The Tsne plot is not clear the movements in color. Its hard to see what are those that are more challenging.
An pipeline figure that summarises the methods employed would be interesting to see.
Is not clear if all the evaluated models have backbones from imagine or Hornet.
The document organisation is ok, follows a general organisation, the language is norma.
Description of actions corroborated with model performance and production would be ok to see, not just number in a confusion matrix, change this to the actions to become more clear to see what are those more challenging.
English is ok; minor typos are easily fixed.
Reduce the amount of text and replace it with some figures that summarize or highlight model, pipeline, etc
Reviewer 4 Report
The authors investigated a research gap in the field of skeleton-based human action recognition and highlighted the need for recognizing small-scale fine-grained human actions using deep learning methods. They proposed a novel approach that utilizes heatmap-based pseudo videos and a unified model applicable to all modality datasets. In my opinion, the manuscript is interesting but there are some concerns that need to be addressed:
1- The computation time of the training phase of the proposed method and the considered methods for comparison should be report in a separate table. Additionally, the efficiency of the method should be discussed in relation to other methods.
2- Two datasets may not be sufficient to determine the effectiveness of the method. I recommend adding additional datasets to comprehensively verify the method.
1- the computetion time of the training phase of the proposed method and considered methods for comparison should be reported in seprate table and accordingly the efficenty of the method should be reported and discussed rather than others.
2- two datsaets is not enough to dicide the method is good or not, I recommended to add a additional datasets to verify comperhensively method.
Round 2
Reviewer 1 Report
In the new version of the paper, the author addressed all of the issues raised in my 1st review. So I advise to accept this paper.
In the new version of the paper, the author addressed all of the issues raised in my 1st review. So I advise to accept this paper.
Reviewer 2 Report
1. The clarity of Figure 4 (b) (c) needs to be improved.
2. The author should make minor editing to English expressions
Minor editing of the English language required
Reviewer 3 Report
Alttrhough author have responded my questions, there are still some open issues.
First, regarding TSNE, labels of the colours dost/clusters should be included.
Second, I forgot to mention previously how the proposed heatmap compares to alphapose? There is no mention of a well know recent Sota work that addresses similar problems..
English minor edits easy corrected with online tool
